# From Stochastic Planning to Marginal MAP

**Hao Cui**
Department of Computer Science
Tufts University
Medford, MA 02155, USA
hao.cui@tufts.edu

**Radu Marinescu**
IBM Research
Dublin, Ireland
radu.marinescu@ie.ibm.com

**Roni Khardon**
Department of Computer Science
Indiana University
Bloomington, IN, USA
rkhardon@iu.edu

## Abstract

It is well known that the problems of stochastic planning and probabilistic inference are closely related. This paper makes two contributions in this context. The first is to provide an analysis of the recently developed SOGBOFA heuristic planning algorithm that was shown to be effective for problems with large factored state and action spaces. It is shown that SOGBOFA can be seen as a specialized inference algorithm that computes its solutions through a combination of a symbolic variant of belief propagation and gradient ascent. The second contribution is a new solver for Marginal MAP (MMAP) inference. We introduce a new reduction from MMAP to maximum expected utility problems which are suitable for the symbolic computation in SOGBOFA. This yields a novel algebraic gradient-based solver (AGS) for MMAP. An experimental evaluation illustrates the potential of AGS in solving difficult MMAP problems.

## 1 Introduction

The connection between planning and inference is well known. Over the last decade multiple authors have introduced explicit reductions showing how stochastic planning can be solved using probabilistic inference (for example, [4, 25, 5, 17, 23, 12, 8, 19, 26, 10, 18]) with applications in robotics, scheduling and environmental problems. However, heuristic methods and search are still the best performing approaches for planning in large combinatorial state and action spaces [9, 7, 2].

This paper makes two contributions in this context. We first analyze a recent heuristic planning algorithm that was shown to be effective in practice. SOGBOFA [2] builds an approximate algebraic computation graph capturing marginals of state and reward variables under independence assumptions. It then uses automatic differentiation [6] and gradient based search to optimize action choice. Our analysis shows that the value computed by SOGBOFA's computation graph is identical to the solution of Belief Propagation (BP) when conditioned on actions. This provides an explicit connection between heuristic planning algorithms and approximate inference. Inference through algebraic expressions has been explored before [16] and even applied to planning but both the symbolic representation and algorithms are different from the ones in SOGBOFA.

Our second contribution is in showing how planning algorithms can be used to solve inference problems, making use of the correspondence in the reverse direction from prior work. The original construction for SOGBOFA can be seen to solve Maximum expected Utility (MEU) problems with

decision variables as roots of the corresponding graphical model and one leaf node representing the value which is being optimized. This corresponds to MMAP problems with MAP variables at the roots and a single evidence node at a leaf. We provide a new reduction from MMAP problems to MEU whose output satisfies these requirements. When combined with the SOGBOFA solver this provides a novel inference algorithm, algebraic gradient-based solver (AGS), that can solve general MMAP problems. AGS effectively uses a symbolic variant of BP with gradient search. AGS provides an alternative to the mixed-product BP algorithm of [13] and the stochastic local search algorithm of [20]. An experimental evaluation compares AGS to state of the art algorithms for MMAP [14] and illustrates its potential in solving hard inference problems.

## 2  Preliminaries

**Belief Propagation in Bayesian Networks:** For our results it is convenient to refer to the BP algorithm for directed graphs [21]. A Bayesian Network (BN) is given by a directed acyclic graph where each node $x$ is associated with a random variable and a corresponding conditional probability table (CPT) capturing $p(x|parents(x))$. The joint probability of all variables is given by $\prod_i p(x_i|parents(x_i))$. In this paper we assume that all random variables are binary.

Assume first that the directed graph is a polytree (no underlying undirected cycles). For node $x$, BP calculates an approximation of $p(x|e)$, which we denote by $BEL(x)$, where $e$ is the total evidence in the graph. Let $\pi(x) \equiv p(x|e^+)$ and $\lambda(x) \equiv p(e^-|x)$, where $e^+, e^-$ are evidence nodes reachable from $x$ through its parents and children respectively. We use $\alpha$ to represent a normalization constant and $\beta$ to denote some constant. For a polytree, $x$ separates its parents from its children and we have

$$BEL(x) = \alpha\pi(x)\lambda(x) \tag{1}$$

$$\lambda(x) = \prod_{j \in children(x)} \lambda_{z_j}(x) \tag{2}$$

$$\pi(x) = \sum_w p(x|w) \prod_{k \in parents(x)} \pi_x(w_k) \tag{3}$$

where $\lambda()$ and $\pi()$ incorporate evidence through children and parents respectively. In (3) the sum variable $w$ ranges over all assignments to the parents of $x$ and $w_k$ is the induced value to the $k$th parent. $\lambda_{z_j}(x)$ is the message that a child $z_j$ sends to its parent $x$ and $\pi_x(w_k)$ is the message that a parent $w_k$ sends to $x$. The messages are given by

$$\lambda_x(w_i) = \beta \sum_x \lambda(x) \sum_{w_k:k\neq i} p(x|w) \prod_{k\neq i} \pi_x(w_k) \tag{4}$$

$$\pi_{z_j}(x) = \alpha \prod_{k\neq j} \lambda_{z_k}(x)\pi(x) \tag{5}$$

where in (4) $w_i$ is fixed and the the sum is over values to other parents $w_k$. Since the nodes are binary the messages have two values (i.e., $\lambda_x(w=0)$ and $\lambda_x(w=1)$). The algorithm is initialized by forcing $\pi$ and $\lambda$ of evidence nodes to agree with the evidence, setting $\pi$ of root nodes equal to the prior probability, and setting $\lambda$ of leaves to (1,1), i.e., an uninformative value. A node can send a message along an edge if all messages from its other edges have been received. If the graph is a polytree then two passes of messages on the graph yield $BEL(x) = p(x)$ for all $x$ [21].

The loopy BP algorithm applies the same updates even if the graph is not a polytree. In this case we initialize all messages to (1,1) and follow the same update rules for messages according to some schedule. The algorithm is not guaranteed to converge but it often does and it is known to perform well in many cases. The following property of BP is well known:

**Lemma 1.** *If loopy BP is applied to a BN with no evidence, i.e., $\lambda(x) = (1,1)$ for all $x$ at initialization, then for any order of message updates and at any time in the execution of loopy BP, for any node $x$, $\lambda(x) \propto (1,1)$ and $\lambda_x(w)$ is a constant independent of $w$ for any parent of $x$. In addition, a single pass upadting $\pi$ messages in topological order converges to the final output of BP.*

*Proof.* We prove the claim by induction. Assume that $\lambda(x) = (o,o)$, for some value $o$, and consider the next $\lambda$ message from $x$. From Eq (4) we have

$$\lambda_x(w_i) = \beta \sum_x \lambda(x) \sum_{w_k:k\neq i} p(x|w) \prod_{k\neq i} \pi_x(w_k) = \beta o \sum_{w_k:k\neq i} \prod_{k\neq i} \pi_x(w_k) \sum_x p(x|w) = \beta o$$

where to get the second equality we extract the constant $\lambda(x) = o$ and reorder the summations. The last equality is true because $\sum_x p(x|w) = 1$ and $\sum_{w_k:k \neq i} \prod_{k \neq i} \pi_x(w_k) = 1$. Therefore, $\lambda_x(w_i)$ is a constant independent of $w_i$. Now from Eq 2 we see that $\lambda(w_i) = (1,1)$ as well, and from Eq 5 and 3 we see that it suffices to update $\pi$ messages in topological order. □

**ARollout and SOGBOFA:** Stochastic planning is defined using Markov decision processes [22]. A MDP [22] is specified by $\{\mathbb{S}, \mathbb{A}, T, R, \gamma\}$, where $\mathbb{S}$ is a finite state space, $\mathbb{A}$ is a finite action space, $T(s, a, s') = p(s'|s, a)$ defines the transition probabilities, $R(s, a)$ is the immediate reward of taking action $a$ in state $s$, and $\gamma$ is the discount factor. A policy $\pi : \mathbb{S} \to \mathbb{A}$ is a mapping from states to actions, indicating which action to choose at each state. Given a policy $\pi$, the value function $V^\pi(s)$ is the expected discounted total reward $E[\sum_i \gamma^i R(s_i, \pi(s_i)) \mid \pi]$, where $s_i$ is the $i^{th}$ state visited by following $\pi$ (and $s_0 = s$). The action-value function $Q^\pi : \mathbb{S} \times \mathbb{A} \to \mathcal{R}$ is the expected discounted total reward when taking action $a$ at state $s$ and following $\pi$ thereafter. In this paper we consider finite horizon planning where the trajectories are taken to a fixed horizon $h$ and $\gamma = 1$, i.e., no discounting is used.

In factored spaces [1] the state is specified by a set of variables and the number of states is exponential in the number of variables. Similarly in factored action spaces an action is specified by a set of variables. We assume that all state and action variables are binary. Finite horison planning can be captured using a dynamic Bayesian network (DBN) where state and action variables at each time step are represented explicitly and the CPTs of variables are given by the transition probabilities. In off-line planning the task is to compute a policy that optimizes the long term reward. In contrast, in on-line planning we are given a fixed limited time $t$ per step and cannot compute a policy in advance. Instead, given the current state, the algorithm must decide on the next action within time $t$. Then the action is performed, a transition and reward are observed and the algorithm is presented with the next state. This process repeats and the long term performance of the algorithm is evaluated. On-line planning has been the standard evaluation method in recent planning competitions.

AROLLOUT and SOGBOFA perform on-line planning by estimating the value of initial actions at the current state $s$, $Q^\pi(s, a)$, where a fixed rollout policy $\pi$, typically a random policy, is used in future steps. The AROLLOUT algorithm [3] introduced the idea of algebraic simulation to estimate values but optimized over actions by enumeration. Then [2] showed how algebraic rollouts can be computed symbolically and that the optimization can be done using automatic differentiation [6]. We next review these algorithms. Finite horizon planning can be translated from a high level language (e.g., RDDL [24]) to a dynamic Bayesian network. AROLLOUT transforms the CPT of a node $x$ into a disjoint sum form. In particular, the CPT for $x$ is represented in the form if$(c_{11}|c_{12}...)$ *then* $p_1$ ... if$(c_{n1}|c_{n2}...)$ *then* $p_n$, where $p_i$ is $p(x{=}1)$ and the $c_{ij}$ are conjunctions of parent values which are are mutually exclusive and exhaustive. In this notation $c_{ij}$ is a set of conjunctions having the same conditional probability $p(x{=}1|c){=}p_i$. The algorithm then performs a forward pass calculating $\hat{p}(x)$, an approximation of the true marginal $p(x)$, for any node $x$ in the graph. $\hat{p}(x)$ is calculated as a function of $\hat{p}(c_{ij})$, an estimate of the probability that $c_{ij}$ is true, which assumes the parents are independent. This is done using the following equations where nodes are processed in the topological order of the graph:

$$\hat{p}(x) = \sum_{ij} p(x|c_{ij})\hat{p}(c_{ij}) = \sum_{ij} p_i\hat{p}(c_{ij}) \tag{6}$$

$$\hat{p}(c_{ij}) = \prod_{w_k \in c_{ij}} \hat{p}(w_k) \prod_{\bar{w}_k \in c_{ij}} (1 - \hat{p}(w_k)). \tag{7}$$

The following example from [2] illustrates AROLLOUT and SOGBOFA . The problem has three state variables s(1), s(2) and s(3), and three action variables a(1), a(2), a(3) respectively. In addition we have two intermediate variables $cond1$ and $cond2$ which are not part of the state. The transitions and reward are given by the following RDDL expressions where primed variants of variables represent the value of the variable after performing the action.

```
cond1 = Bernoulli(0.7)
cond2 = Bernoulli(0.5)
s'(1) = if (cond1)  then ˜a(3)  else false
s'(2) = if (s(1))   then a(2)   else false
s'(3) = if (cond2)  then s(2)   else false
reward = s(1) + s(2) + s(3)
```

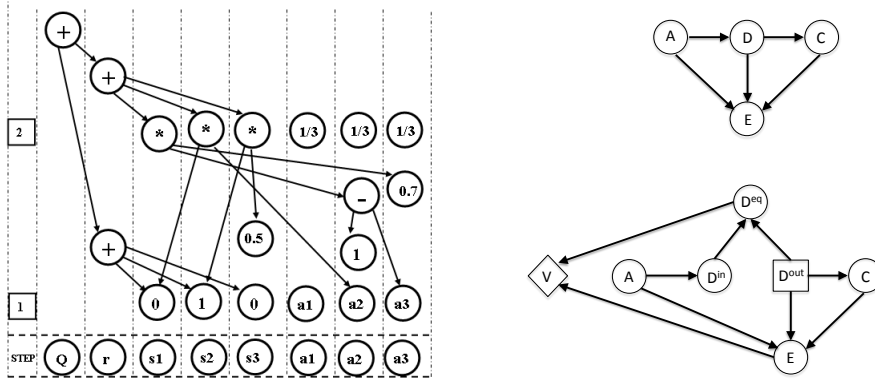

Figure 1: Left: example of SOGBOFA graph construction. Right: Example of reduction from MMAP to MEU. Original graph (top) and transformed graph (bottom).

AROLLOUT translates the RDDL code into algebraic expressions using standard transformations from a logical to a numerical representation. In our example this yields:

```
s'(1) = (1-a(3))*0.7
s'(2) = s(1)*a(2)
s'(3) = s(2) * 0.5
r = s(1) + s(2) + s(3)
```

These expressions are used to calculate an approximation of marginal distributions over state and reward variables. The distribution at each time step is approximated using a product distribution over the state variables. To illustrate, assume that the state is $s_0 = \{s(1)=0, s(2)=1, s(3)=0\}$ which we take to be a product of marginals. At the first step AROLLOUT uses a concrete action, for example $a_0 = \{a(1)=1, a(2)=0, a(3)=0\}$. This gives values for the reward $r_0 = 0 + 1 + 0 = 1$ and state variables $s_1 = \{s(1)=(1-0)*0.7=0.7, s(2)=0*0=0, s(3)=1*0.5 = 0.5\}$. In future steps it calculates marginals for the action variables and uses them in a similar manner. For example if $a_1 = \{a(1)=0.33, a(2)=0.33, a(3)=0.33\}$ we get $r_1 = 0.7+0+0.5 = 1.2$ and $s_2 = \{s(1)=(1-0.33)*0.7, s(2)= 0.7 * 0.33, s(3)=0*0.5\}$. Summing the rewards from all steps gives an estimate of the $Q$ value for $a_0$. AROLLOUT randomly enumerates values for $a_0$ and selects the one with the highest estimate.

The main observation in SOGBOFA is that instead of calculating numerical values, as illustrated in the example, we can use the expressions computing these values to construct an explicit directed acyclic graph representing the computation steps, where the last node represents the expectation of the cumulative reward. SOGBOFA uses a symbolic representation for the first action and assumes that the rollout uses the random policy. In our example if the action variables are mutually exclusive (such constraints are often imposed in high level domain descriptions) this gives marginals of $a_1 = \{a(1)=0.33, a(2)=0.33, a(3)=0.33\}$ over these variables. The SOGBOFA graph for our example expanded to depth 1 is shown In Figure 1. The bottom layer represents the current state and action variables. Each node at the next level represents the expression that AROLLOUT would have calculated for that marginal. To expand the planning horizon we simply duplicate the second layer construction multiple times.

Now, given concrete marginal for the action variables at the first step, i.e., $a_0$, one can plug in that value into the computation graph and compute the value of the final $Q$ node. This captures the same calculation as AROLLOUT. In addition, the explicit graph allows us to compute the gradients of the $Q$ value with respect to the action variables, using automatic differentiation. We refer the reader to [6] for details on automatic differentiation; the basic idea is similar to backpropagation of gradients in neural network learning which can be generalized to arbitrary graphs. In this manner we can perform gradient search over marginals for action variables in $a_0$ and effectively select values for the action variables at the first step. SOGBOFA includes several additional heuristics including dynamic control of simulation depth, dynamic selection of gradient step size, maintaining domain constraints, and a balance between gradient search and random restarts. Most of this is orthogonal to the topic of this paper and we omit the details.

## 3   AROLLOUT **is Equivalent to BP**

We first show that the computation of AROLLOUT can be rewritten as a sum over assignments.

**Lemma 2.** AROLLOUT*'s calculation in Eq (6) and (7) is equivalent to (8) where $W$ is the set of assignment to the parents of $x$.*

$$\hat{p}(x) = \sum_W p(x|W) \prod_{k \in parents(x)} \hat{p}(w_k)^{w_k}(1 - \hat{p}(w_k))^{1-w_k}. \tag{8}$$

*Proof.* The sum in (8) can be divided into disjoint sets of assignments according to the $c_{ij}$ they satisfy. Consider one fixed $c_{ij}$. Let $w_l$ be a parent of $x$ which is not in $c_{ij}$. Let $W(c_{ij})$ be the assignments to the parents of $x$ which satisfy $c_{ij}$, and $W_{\setminus w_l}(c_{ij})$ be the assignments to the parents of $x$ except for $w_l$ which satisfy $c_{ij}$. Since $w_l$ is not in $c_{ij}$, $W_{\setminus w_l}(c_{ij})$ is well defined. We have that

$$\sum_{W \in W(c_{ij})} p(x|W) \prod_k \hat{p}(w_k)^{w_k}(1 - \hat{p}(w_k))^{1-w_k} \tag{9}$$

is equal to $\sum_{W_{\setminus w_l}(c_{ij})} \left( p(x|W, w_l = 1)\hat{p}(w_l) \prod_{k \neq l} \hat{p}(w_k)^{w_k}(1 - \hat{p}(w_k))^{1-w_k} + p(x|W, w_l = 0)(1 - \hat{p}(w_l)) \prod_{k \neq l} \hat{p}(w_k)^{w_k}(1 - \hat{p}(w_k))^{1-w_k} \right)$. Now since $w_l$ is not in $c_{ij}$ the assignment of $w_l$ does not affect the probability of $x$. So for $W \in W_{\setminus w_l}(c_{ij})$ we have $p(x|W, w_l{=}1) = p(x|W, w_l{=}0) = p(x|W)$ and therefore the above sum can be simplified to

$$\sum_{W_{\setminus w_l}(c_{ij})} p(x|W) \prod_{k \neq l} \hat{p}(w_k)^{w_k}(1 - \hat{p}(w_k))^{1-w_k}.$$

Applying the same reasoning to all individual $w_l \notin c_{ij}$, we get that Eq (9) is equal to

$$\sum_{W \in w_p(c_{ij})} p(x|W) \prod_{k \in c_{ij}} \hat{p}(w_k)^{w_k}(1 - \hat{p}(w_k))^{1-w_k}$$

where $w_p(c_{ij})$ is the set of assignments to variables in $c_{ij}$ which satisfy $c_{ij}$, that is, we removed all parents not in $c_{ij}$ from the expression. Now, because $c_{ij}$ is a conjunction, $w_p(c_{ij})$ includes a single assignment where if $w_k \in c_{ij}$ we have $w_k = 1$ and if $\bar{w}_k \in c_{ij}$ we have $w_k = 0$. In addition, for this assignment $W$ we have that $p(x = 1|W) = p_i$. Therefore, the last expression simplifies to

$$p_i \prod_{w_k \in c_{ij}} \hat{p}(w_k) \prod_{\bar{w}_k \in c_{ij}} (1 - \hat{p}(w_k)).$$

Finally, because the $c_{ij}$ are mutually exclusive and exhaustive, the sum over the disjoint sets of assignments is identical to the sum in (6). $\qquad\square$

**Proposition 3.** *The marginals calculated by* AROLLOUT *are identical to the marginals calculated by BP on the DBN generated by the planning problem, conditioned on the initial state, initial action and rollout policy, and with no evidence.*

*Proof.* By Lemma 1, $\lambda(x)$ and $\lambda_x(w_i)$ are $\propto (1, 1)$ for all nodes. Therefore, backward messages do not affect the result of BP and we can argue inductively going forward from roots to leaves in the DBN. By Eq (1) and Lemma 1 we have $BEL(x) = \alpha\pi(x)\lambda(x) = \pi(x)$ where the last equality is true because $\pi(x)$ is always normalized. Therefore from Eq (3) we have

$$BEL(x) = \sum_w p(x|w) \prod_k \pi_x(w_k). \tag{10}$$

Now, from Eq (5) and Lemma 1 we have

$$\pi_x(w_k) = \pi(w_k) = BEL(w_k) \tag{11}$$

and substituting (11) into (10) we get

$$BEL(x) = \sum_w p(x|w) \prod_k BEL(w_k). \tag{12}$$

Inductively assuming $BEL(w_k = 1) = \hat{p}(w_k)$ and $BEL(w_k = 0) = 1 - \hat{p}(w_k)$, we can rewrite (12) as $BEL(x = 1) = \hat{p}(x) = \sum_W p(x|W) \prod_k \hat{p}(w_k)^{w_k}(1 - \hat{p}(w_k))^{1-w_k}$, which is identical to (8). $\quad\square$

# 4   Algebraic Solver for Marginal MAP

Marginal MAP [20, 13, 11, 14] is a complex inference problem seeking a configuration of a subset of variables that maximizes their marginal probability. Recall that the graph construction in SOG-BOFA evaluates exactly to the value returned by AROLLOUT . Therefore, the result in the previous section shows that SOGBOFA can be understood as using gradient search for the best action where the evaluation criterion is given using BP but calculated symbolically. In this section, we show that this approach can be used for MMAP yielding a novel solver for these problems.

The input to a MMAP problem is a Bayesian network $G$ where the nodes in the network are divided into 3 sets $E, D, S$ standing for evidence nodes, MAP (or decision) nodes and sum nodes, with a specification of values to evidence nodes $E = e$. The goal is to find $argmax_{D=d} \sum_{S=s} p(D = d, S = s, E = e)$. Anytime algorithms are typically scored using the log of marginal probability; the score for solution $D = d$ is $Q = \log \sum_{S=s} p(D = d, S = s, E = e)$. Current state of the art exact algorithms use branch and bound techniques (e.g., [11]). Various approximation algorithms for MMAP exist including mixed product belief propagation [13], an extension of BP that directly addresses MMAP and is therefore closely related to the algorithms in this paper.

To make the connection more precise we show that the optimization problem in SOGBOFA is a maximum expected utility (MEU) problem. In the graphical models literature such problems are formalized using influence diagrams (ID). An influence diagram is a Bayesian network where two additional types of nodes are allowed in addition to random variable nodes $R$. Decision nodes $D$ represent variables whose values (conditioned on parents) are being optimized, and value nodes $V$ are leaves. The IDs that arise in this paper (arising from SOGBOFA and later as the output of our reduction) satisfy additional syntactic constraints: decision nodes do not have parents and there is a single value node $V$. We restrict the discussion to such IDs. This avoids subtleties in defining the optimization problem. In this case, given an ID the goal is to find $argmax_{D=d} E_{R=r}(V|D = d) = argmax_{D=d} \sum_{R=r} \sum_{V=v} v \cdot p(V = v, R = r|D = d)$.

Consider SOGBOFA with a fixed rollout policy $\pi$ and w.l.o.g. assume a single binary node $V$ representing cumulative reward.[1] For a start action $a$ we have $Q = p(V = 1|a, \pi) = E(V|a, \pi)$. Now assuming a uniform prior over $a$ we have $p(V = 1|a, \pi) \propto p(V = 1, a|\pi)$ and $\arg\max_a p(V = 1|a, \pi) = \arg\max_a p(V = 1, a|\pi) = \arg\max_a \sum_S p(V = 1, a, S|\pi)$ where $S$ are the state variables. It is obvious from the last expression that SOGBOFA can be seen to solve MEU for the restricted class of IDs and that this is the same as a restricted version of MMAP problems, where the structural constraints on $a$ and $V$ are given above. This implies:

**Corollary 4.** SOGBOFA *can be directly used to solve Marginal MAP problem on graphs with parentless MAP nodes and only one evidence node at a leaf.*

The question is whether we can use SOGBOFA for general MMAP problems. We next show how this can be done. Mauá [15] gave a reduction from MMAP to maximum expected utility (MEU) in influence diagrams (reduction 5.1) which satisfies our syntactic requirements. The reduction preserves correctness under exact inference. However, with that construction there is no direct forward path that connects decision nodes, downward evidence nodes, and the value node. Recall that SOGBOFA uses forward propagation of marginals in the directed graph. If no such path exists then downward evidence is ignored and the result of forward BP inference is not informative. We give a new reduction that avoids this issue and in this way introduce a new algorithm for MMAP.

**Reduction:** Let the input for the MMAP problem be given by $E, D, S$ as above. Without loss of generality we may assume that each $E_i \in E$ does not have any children. If it does we can first disconnect the edges to its children and substitute the value of the evidence node directly into the child's CPT. The reduction performs two modifications on the original graph. (1) Each MAP node $D_i \in D$ is replaced by three nodes: $D_i^{in}$, $D_i^{out}$ and $D_i^{eq}$. $D_i^{in}$ has all inputs of $D_i$ and the same CPT. $D_i^{out}$ has no parents and it connects to all the outputs of $D_i$. Finally both $D_i^{in}$ and $D_i^{out}$ are parents of $D_i^{eq}$ and the CPT for $D_i^{eq}$ specifies that $D_i^{eq}$ is true iff $D_i^{in} = D_i^{out}$. (2) We add a new leaf utility node $V$ whose CPT captures a logical conjunction requiring that all evidence nodes have their observed values and that all nodes $D_i^{eq}$ are true. Although $V$ may have many parents we can represent its CPT symbolically and this does not adversely affect the complexity of our

algorithm. The CPTs for all other nodes are unchanged except that a parent $D_i$ is changed to $D_i^{out}$. The influence diagram problem is to find the setting for variables in $\{D_i^{out}\}$ which maximize the expected utility $E[V|\{D_i^{out}\}] = p(V = 1|\{D_i^{out}\})$. An example of this construction with one evidence node $E$ and one MAP node $D$ is shown in Figure 1. We have:

**Proposition 5.** *Let $G_1$ represent the original MMAP problem, $G_2$ the transformation into MEU, and let $E=e$ be the evidence for the MMAP problem and $D=d$ an assignment to the MAP variables. Then, $p_{G_1}(D=d, E=e) = E_{G_2}[V|D^{out}=d]$.*

*Proof.* (sketch) We illustrate how the claim can be proved for the example from Figure 1. In this case, $p(D=d, E=e) = \sum_A \sum_C p(A)\, p(D=d|A)\, p(C|D=d)\, p(E=e|A, D=d, C)$. Now in $G_2$,

$$E[V|D^{out}=d] = p(D^{eq}=1, E=e|D^{out}=d)$$
$$= \sum_A \sum_C \sum_{D^{in}} p(D^{eq}=1|D^{in}, D^{out}=d)p(A)\, p(D^{in}|A)\, p(C|D^{out}=d)\, p(E=e|A, D^{out}=d, C).$$

Now replace the sum over $D^{in} \in \{0,1\}$ with a sum over the cases $D^{in}=D^{out}=d$ and $D^{in} \neq D^{out}$ and observe that $p(D^{eq}=1|D^{in}, D^{out}=d)$ is 1 in the first case and 0 in the second. Therefore the last expression can be simplified to

$$\sum_A \sum_C p(A)\, p(D^{in}=d|A)\, p(C|D^{out}=d)\, p(E=e|A, D^{out}=d, C)$$

which by construction is identical to the value for $G_1$.

The proof for the general case follows along the same steps. The crucial point is to replace the sum over $D_i^{in}$ into the cases where it is the same vs. not equal to $D_i^{out}$. This shows that the irrelevant terms cancel out and the remaining terms are identical to the original ones. $\square$

The reduction allows us to solve general MMAP problems using the SOGBOFA heuristic:

**AGS – Algebraic Gradient Based Solver for MMAP:**

1. Given a MMAP problem $G_1$ with evidence $E = e$, decision nodes $D$ and sum nodes $S$ use the reduction to obtain a MEU problem $G_2$ with utility node $V$ and decision nodes $D^{out}$.

2. Generate the SOGBOFA graph $G_{SOG}$ from the MEU problem where decision nodes are treated as action nodes and $V$ is the $Q$ node of the planning problem.

3. Use the gradient based optimizer in SOGBOFA (gradient ascent with random restarts) to optimize the marginal probabilities of variables $\{D_i^{out}\}$.

4. Extract a discrete solution from the marginal probabilities by thresholding: $D_i^{out} = 1$ iff $p(D_i^{out} \geq 0.5)$.

**Corollary 6.** *AGS can be used to solve general Marginal MAP problems.*

## 5  Experimental Validation

In this section, we explore the potential of AGS in solving complex MMAP problems. Specifically, we evaluate the anytime performance of AGS and two natural baselines. The first is the Mixed Product BP (MPBP) algorithm of [13]. MPBP uses belief propagation and is therefore related to AGS, but in MPBP the search over MAP variables is integrated into the messages of BP and like BP it can be derived from the corresponding optimization problem. The second algorithm is the recently developed Alternating best-first with depth-first AND/OR search (AAOBF) [14]. AAOBF interleaves best-first and depth-first search over an AND/OR search space to compute both anytime solutions (corresponding to lower bounds) as well as upper bounds on the optimal MMAP value. AAOBF was shown to have excellent anytime performance and dominate other algorithms.

For the evaluation we use several problems from the UAI competition 2008. The original challenge problems were given for sum inference, specifying the network and evidence nodes. Following previous work, we use these for MMAP by selecting a subset of the variables as MAP nodes. To explore the performance of the algorithms we vary the proportion of MAP variables in each instance, and for each fixed ratio we generate 20 MMAP problems by picking the MAP nodes at random.

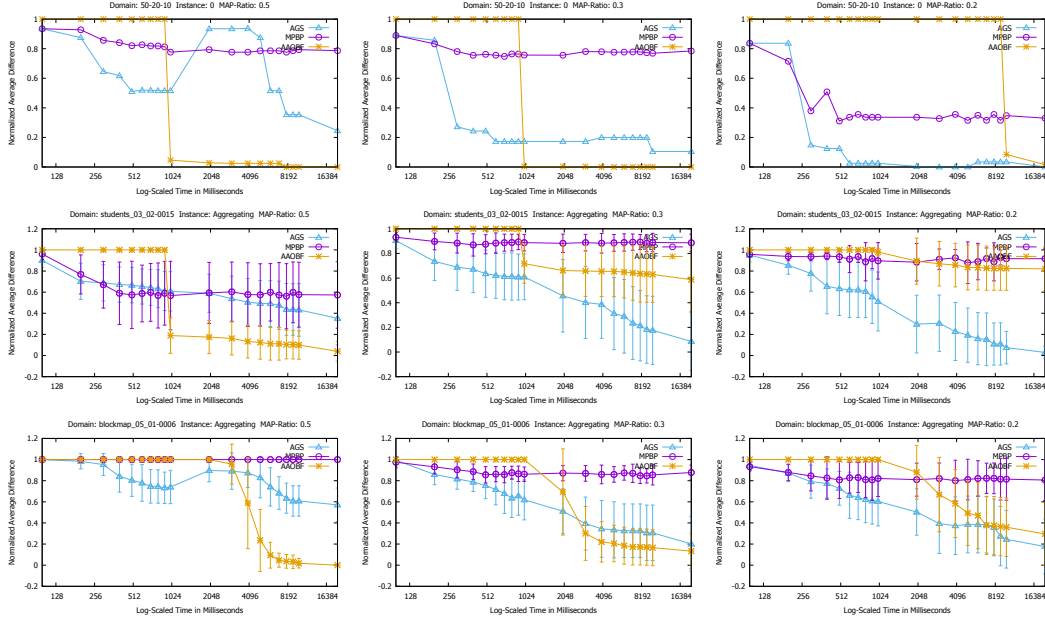

Figure 2: Experimental results on UAI instances. Each row shows results for one instance. The top row shows results of one run. Other rows show results aggregated over 20 random choices of MAP nodes. The columns correspond to proportion of MAP variables (0.5, 0.3, 0.2).

For AAOBF we use the implementation of [14] that can process the UAI competition problems directly. AGS requires CPTs as expressions and our implementation extracts such expressions from the tabular representation of the UAI problems as a preprocessing step. This is not computationally demanding because the tabular representation is naturally restricted to have a small number of parents. We use our own implementation of MPBP, and for consistency the MPBP implementation benefits from the same expression representation of CPTs as AGS. More specifically, we use the join graph version of MPBP (algorithm 5 of [13]) and run it on the factor graph which is obtained from the BN. Since the factor graph is not a cluster *tree* we are running loopy MPBP. The max clusters of MPBP correspond to individual MAP variables, and sum nodes include both individual sum variables and factors in the original BN. Factor nodes and sum nodes perform the same computations as in standard loopy BP. The Max cluster with node $i$ calculates a message to factor $j$ as follows: first calculate the product of all incoming messages from factors other than $j$. Then, noting that we have binary variables and thus only two entries in a message, zero out the smaller entry if it is strictly smaller. MPBP keeps iterating over updates to nodes until it runs out of time or the maximal change of the messages becomes smaller than 0.0001. While [13] introduce annealing and restarts to improve the performance of this algorithm we do not use them here. Note that MPBP can get into a "contradiction state" when the graph has logical constraints, i.e., messages can become (0,0) or legal states are ruled out. AGS does not suffer from this problem. However, to enable the comparison we modified the UAI instances changing any 0 probability to 0.0001 (and 1 to 0.9999). The implementation of AAOBF replaces every 0 with 0.000001 for similar reasons. The solutions of all algorithms are evaluated off line using an exact solver which uses the same code base as AAOBF.

Figure 2 shows the results. Each algorithm is scored using log marginal probability. The plot shows a relative score $c_t = \frac{a_t - b}{a_t}$ where $a_t$ is the score of algorithm $a$ at time $t$ and $b$ is the best score found by any of the algorithms for this instance. This guarantees that relative scores are between 0 and 1, where the best value is 0. When an algorithm finds an inconsistent solution (probability 0) or does not find a solution we replace the score with 1. We show results for 3 problems, where for one problem (top row) we show results for a single run and for two problems we show results aggregated over 20 runs. Comprehensive results with individual runs and aggregated runs on more instances are given in the supplementary material. Each column in Figure 2 corresponds to a different proportion of MAP variables in the instance (0.5,0.3,0.2 respectively). The results for individual runs show more clearly transitions between no solution and the first solution for an algorithm whereas this is averaged in aggregate results. But the trends are consistent across these graphs. We see that AAOBF

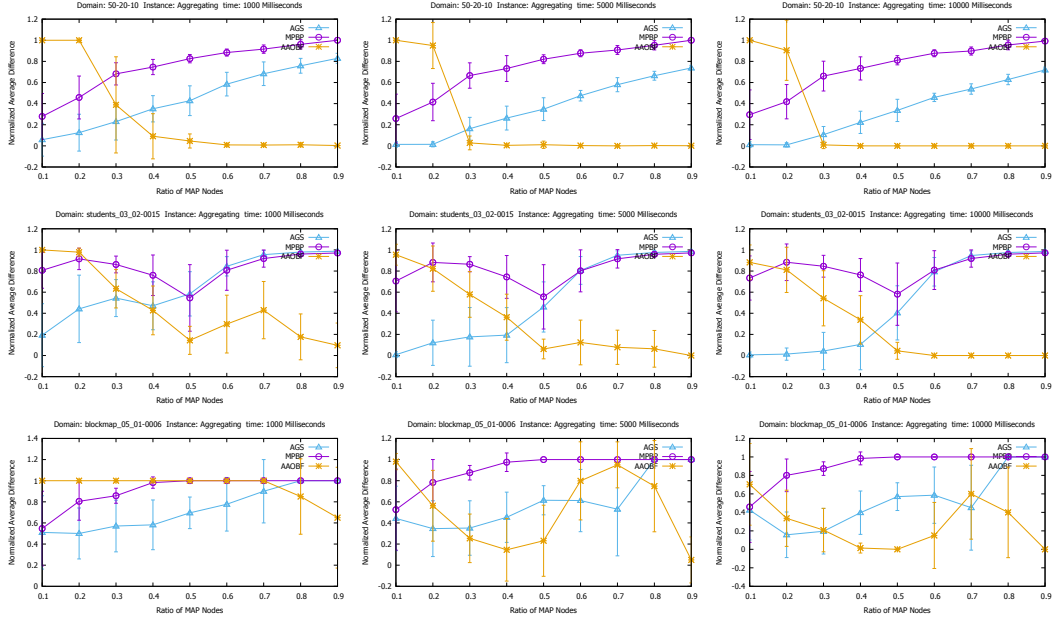

Figure 3: Experimental results on different proportion of MAP variables. Each row corresponds to one problem. Each column corresponds to different running time, from left to right 1, 5 and 10 seconds.

has a larger initial overhead and AGS and MPBP are faster to find the first solutions and that AGS performs better than MPBP. AAOBF is significantly affected by the complexity of the conditional sum inference problems (i.e., evaluating the score of a specific MAP assignment). For the problems with 50% of MAP variables (and only 50% sum variables) the complexity is not too high and the search successfully finds high quality solutions. For these problems AAOBF dominates both AGS and MPBP. On the other hand, with 70% and 80% of sum variables the summation problems are harder and AAOBF is slower to find solutions. In this case AGS dominates as it finds reasonable solutions fast and improves with time. To further illustrate the impact of summation difficulty we run the algorithms in the same setup but with a fixed bound on run time varying the proportion of MAP variables from 0.1 to 0.9. Figure 3 shows results for the same 3 problems averaged over 20 runs, for run time of 1,5,10 seconds in corresponding columns. Here, we clearly see the transition in relative performance of the algorithms as a function of the proposition of MAP variables. We also see that with shorter run time AGS dominates for a larger range of problems. To summarise, given enough time AAOBF will find an optimal solution and can dominate AGS which is limited by the approximation inherent in BP. However, with a limited time and difficult conditional summation problems AGS provides a better tradeoff in finding solutions quickly.

## 6 Conclusions

The paper identifies a connection between a successful heuristic for planning in large factored spaces and belief propagation. The SOGBOFA heuristic performs its estimation symbolically and through that performs its search using gradients. This suggests a general scheme for approximate MMAP algorithms where the MAP value is represented using an explicit computation graph which is optimized directly through automatic differentiation. The instantiation of this scheme in AGS shows that it improves over the anytime performance of state of the art algorithms on problems with hard summation sub-problems. In addition, while previous work has shown how inference can be used for planning, this paper shows how ideas from planning can be used for inference. We believe that these connections can be further explored to yield improvements in both fields.

### Acknowledgments

This work was partly supported by NSF under grant IIS-1616280. Some of the experiments in this paper were performed on the Tufts Linux Research Cluster supported by Tufts Technology Services.

## Footnotes

[1]There are standard reductions (see [13]) showing how to translate a general reward to a binary variable whose expectations are proportional so that the optimization problems are equivalent.

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
