[Supplementary Material]

# Supplementary Material for: From Stochastic Planning to Marginal MAP

**Hao Cui**
Department of Computer Science
Tufts University
Medford, MA 02155, USA
hao.cui@tufts.edu

**Radu Marinescu**
IBM Research
Dublin, Ireland
radu.marinescu@ie.ibm.com

**Roni Khardon**
Department of Computer Science
Indiana University
Bloomington, IN, USA
rkhardon@iu.edu

The supplement includes the entire set of experimental results. The results include all 6 problems with which we ran experiments. The number of nodes and number of evidence variables for each problem is given in Table 1. Note that we chose problems with a moderate number of variables. This is needed in order to enable an evaluation with an *exact solver*. While the run time of the algorithms in the experiments is short, and AGS does yield solutions for much larger problems, the run time of the evaluation of the solutions with an exact solver becomes too expensive for larger problems.

In the first experiment we fix 3 ratios for MAP nodes (0.5, 0.3, 0.2) and measure performance as a function of run time. For each problem we generate 20 random instances by selecting the MAP nodes at random. We include the normalized difference in log marginal for both aggregating over all instances and for the first individual instance. The plots are given respectively in Figure 1 and Figure 2. There is a clear trend on all problems. Given the same graph, when the ratio of MAP nodes is high (ratio of sum nodes is low) AAOBF dominates the other two algorithms. However, as the ratio of MAP nodes gets lower, and the cost of summation is higher, AGS performs better. This is because AGS' symbolic representation for BP makes the summation efficient, so that it suffers less when the summation gets harder.

Also note that although MPBP and SOGBOFA both uses BP approximation, SOGBFOA seems to have a much more efficient search over the MAP assignments. The performance of MPBP might improve by incorporating random initializations and annealing however this requires some tuning.

In the second experiment we fix 3 values for the run time (1,5,10 seconds) and vary the proportion of MAP variables. Figure 3 shows results for the same problems averaged over 20 runs. Here we clearly see the transition in relative performance of the algorithms as a function of the proposition of MAP variables. Note that for fs-04 the summation problem is relatively easy and even 1 second is sufficient for AAOBF to find a solution (this can be seen in Figure 1) explaining why it dominates for the entire range. However, here too, the relative performance of AGS improves with decreasing ratio of MAP variables, and we expect to see a similar transition if a shorter run time is used. We also see that with shorter run time AGS dominates for a larger range of problems.

To summarize, given enough time AAOBF will find an optimal solution and can dominate AGS which is limited by the approximation inherent in BP. However, with a limited time and difficult conditional summation problems AGS provides a better tradeoff in finding solutions quickly.

| Domain | 50-20-10 | 90-20-1 | blockmap-05-01_0006 | fs-04 | mastermind_03_08_03-0001 | students_03_02-0015 |
|---|---|---|---|---|---|---|
| #nodes | 400 | 400 | 700 | 262 | 1220 | 376 |
| #evidence | 1 | 1 | 46 | 226 | 27 | 2 |

Table 1: Number of nodes and number of evidence for each domain.

Figure 1: Experimental results on UAI instances. Each row shows results for one instance aggregated over 20 random choices of MAP nodes. The columns correspond to proportion of MAP variables (0.5, 0.3, 0.2).

Figure 2: Experimental results on UAI instances. Each row shows results the first random instance. The columns correspond to proportion of MAP variables (0.5, 0.3, 0.2).

Figure 3: Experimental results on different proportion of MAP variables. Each row corresponds to one problem. Each column corresponds to different running time, from left to right 1, 5 and 10 seconds.