[Reviews · NeurIPS 2018]

Reviewer 1



The paper revisits the SOGBOFA heuristic planning algorithm. It shows that SOGBOFA can be viewed as running belief propagation conditions on the actions. Based on this, a novel Marginal MAP (MMAP) solver is presented. The idea is to reduce the MMAP problem to maximum expected utility such that SOGBOFA can be employed for solving the problem. Overall, the direction of using algebraic expressions in planning is of great interest. Moreover, the present paper is the first, to my knowledge, that tackles MMAP using algebraic computations, at least the way presented. However, the paper is missing some related approaches. For instance, Martin Mladenov, Vaishak Belle, Kristian Kersting: The Symbolic Interior Point Method. AAAI 2017: 1199-1205 also makes use of algebraic computations for planning problem. Moreover, interior point solvers have been used (within) MMAP approaches. While not quite there, it is very much related and should be discussed. Also, the authors should clarify the connection to H. Cui and R. Khardon, Lifted Stochastic Planning, Belief Propagation and Marginal MAP, The AAAI-18 Workshop on Planning and Inference, held with the AAAI Conference on Artificial Intelligence (AAAI), 2018 While indeed the NIPS 2018 policy allows resubmissions of workshop papers, this only holds for non-archival workshops. AAAI workshops often have proceedings hosted by AAAI Press. As far as I see it, the present paper goes significantly beyond that paper. In any case, the main contribution, namely the reduction to MMAP to MEU in combination with SOGBOFA is interesting. It is triggered by a known reduction. However, the known reduction had to be modified in a non-trivial way in order to apply to SOGBOFA. This variant of the reduction is interesting, and the empirical results show that the resulting AGS is a valid alternative to SOTA, in particular on hard summation problems. I really enjoyed reading the paper and expect that it will receive a lot of attention.

Reviewer 2



Main ideas ========== The paper develops the relation between solving an MDP and performing inference in a Bayesian network. The direction, however, is novel as far as I can tell: using MDP algorithms to solve an inference problem. The first part shows that an existing MDP algorithm (ARollout) is in fact performing a BP iteration over the DBN that represents the MDP. In the second part, a different MDP algorithm (SOGBOFA) is used to solve a particular inference problem of choosing a subset of values with the maximal marginals (MMAP). The resulting SOGBOFA-based solver often loses to the state-of-the-art, but for harder cases it can outperform the state of the art. The idea is interesting and the results are useful, mainly in expanding the understanding of the relation between planning and inference. Regarding quality, I was not able to verify all details of the proofs and algorithms, but as far as I could verify the paper is technically sound. Strength ======== Originality: I am not aware of work on using planning to solve inference problem, and on mapping an MDP solver to BP. Significance: I like the mapping proposed by the authors, and I wonder if it can be extended to other planning algorithms, i.e., using them to solve MMAP as well. Weaknesses ========== Clarity: the paper was very hard for me to follow. I mention some of the reasons below, but in general, more background is needed. However, I admit that this work is somewhat outside my area of expertise, and so perhaps this background is not needed for someone that is more in the field. Examples of clarity issues: Line 91: "... to estimate values ..." - which values? Line 97: "c_{ij}" what is the j representing? Line 130: "... concrete values, ... use the expressions ..." - which values? which expressions? Line 140: "... an evaluation of the graph ..." - what do you by this? In general the paragraph starting with line 140 should be elaborated. Also, the term "automatic differentiation" is used several times but is not explained (and a reference for it is also not provided). Line 197: "For the purpose of this paper it suffices to consider IDs where decision nodes do not have parents and there is a single value node V." - please better explain: why is it sufficient to consider this? I think that you relax this requirement later in the paper, but if that's the case then say so explicitly, e.g., "We begin by assuming X and then relax it." Rebuttal question: ================= I find the reduction from MMAP to SOGBOFA confusing, since usually reduction is done between problems. This raises the following question: can one reduce an MMAP problem to an MDP problem? I belive the answer is yes, in which case why should one use specifically SOGBOFA to solve this problem? Minors: ====== Line 96: you probably want to have "if" not in math mode, to get the spacing correct. Line 97: "... where p_i is p(x=1) ..." - do you mean "p_1 is p(x=1)"?

Reviewer 3



The authors proposed an interesting approach, where a planning method, SOGBOFA, is used for solving general Marginal MAP (MMAP) problems over a graphical model. The MMAP problem can be NP-hard even on trees, thus relaxations are required in most cases. Then main contribution of the paper is to build a connection between belief propagation and SOGBOFA. Particularly it proves the equivalence between AROLLOUT (a module of SOGBOFA) and belief propagation over bayes networks. The authors show that for particular class of MMAP problem, SOGBOFA can be applied directly, while for general MMAP problem, the authors proposed a reduction algorithm to convert MMAP to maximum expected utility (MEU) problem, and then solve the the problem via SOGBOFA. The paper is mostly well-written, and the theory seems sound and convincing. I only have a bit concern about the technical detail. Comparing to another variational inference based MMAP method, Mixed Product BP (MPBP), the proposed algorithm is based on belief propagation over Bayes networks, while MPBP is based on belief propagation over conditional random fields. In Bayes networks, we need not worry about the partition function since all factors are conditional distributions. However, for conditional random fields (CRF) when doing belief propagating we need take care of the partition function. Thus in BP over CRF we need some normalization factors. In Eq. (1) and (5), there is a normalization factor for Bayes networks, why do we need such a factor? I also would like to know what kind of graphs are used in the experiments, bayes network or CRF? Minor suggestion: Draw Figure 2 with log-scaled time may results in a better look.